# An Evaluation System of Robotic End-Effectors for Food Handling

**DOI:** 10.3390/foods12224062

**Published:** 2023-11-08

**Authors:** Zhe Qiu, Hannibal Paul, Zhongkui Wang, Shinichi Hirai, Sadao Kawamura

**Affiliations:** 1Research Organization of Science and Technology, Ritsumeikan University, Kusatsu 525-0058, Japan; qiuzhe@fc.ritsumei.ac.jp (Z.Q.); hpaul@fc.ritsumei.ac.jp (H.P.); 2Department of Robotics, Ritsumeikan University, Kusatsu 525-0058, Japan; hirai@se.ritsumei.ac.jp; 3Ritsumeikan Global Innovation Research Organization, Ritsumeikan University, Kusatsu 525-0058, Japan; kawamura@se.ritsumei.ac.jp

**Keywords:** robotic end-effector, food property, categorization, performance evaluation

## Abstract

Owing to Japan’s aging society and labor shortages, the food and agricultural industries are facing a significant demand for robotic food handling technologies. Considering the large variety of food products, available robotic end-effectors are limited. Our primary goal is to maximize the applicability of existing end-effectors and efficiently develop novel ones, and therefore, it is necessary to categorize food products and end-effectors from the viewpoint of robotic handling and establish their relationships through an effective evaluation approach. This study proposes a system for evaluating robotic end-effectors to identify appropriate ones and develop new ones. The evaluation system consists of food categorization based on food properties related to robotic handling, categorization of robotic end-effectors based on their grasping principles, a robotic system with visual recognition based on Robot Operating System 2 (ROS 2) to conduct handling tests, a scoring system for performance evaluation, and a visualization approach for presenting the results and comparisons. Based on food categorization, 14 real food items and their corresponding samples were chosen for handling tests. Seven robotic end-effectors, both commercialized and under development, were selected for evaluation. Using the proposed evaluation system, we quantitatively compared the performance of different end-effectors in handling different food items. We also observed differences in the handling of real food items and samples. The overall performance of an end-effector can be visualized and quantitatively evaluated to demonstrate its versatility in handling various food items.

## 1. Introduction

The food industry is a complex and rapidly expanding sector that has been constantly striving to satisfy the demands of population growth and adapt to changing human lifestyles in recent years. Traditionally, the food industry encompasses various activities, including processing, sorting, distribution, preparation, packaging, and other services. Food contamination is another consistent challenge facing the food industry worldwide, as demonstrated by multiple recalls due to poor food quality, undeclared allergens, and other forms of contamination. To overcome global labor shortages and promote food safety in the food industry, the use of robotics and automation has drawn significant attention from both academia and industry [1]. The COVID-19 pandemic has accelerated the adoption of food automation across multiple manufacturing stages in the food industry to ensure food safety and maintain a reliable food supply [2].

Food handling is a fundamental operation in food factories, where pick-and-place operations are commonly employed for packaging food items. Robotic systems have emerged as efficient solutions for such operations because they are adaptable to various food products. However, several challenges continue to hinder the widespread implementation of robotic systems in the food industry: (1) most existing robotic end-effectors have been developed to handle specific food products and have limitations in handling a wide range of food products; (2) the speed of pick-and-place operations performed by robotic systems must be improved to match the efficiency of skilled workers; (3) accurate recognition algorithms are required to effectively identify food items in a bin-picking scenario; and (4) the robotic system has to be food grade, easy to sterilize, easy to use, and, most importantly, it must have a high cost performance. This study focuses on addressing the first challenge regarding robotic end-effectors in food handling and proposes a system for evaluating the performance of different end-effectors to help select proper existing end-effectors and develop new ones. The evaluation system is based on food and end-effector categorizations, experimental tests, and scoring and visualization approaches.

The food industry has a long history and many studies have been conducted on food property estimation and categorization. Among the food properties related to robotic handling, elasticity, which indicates the softness of food products, has been the most frequently studied for applications in food science, mastication, and cooking [3,4,5,6]. Viscoelasticity and rheology, which further consider viscosity and residual deformation, have also been investigated for various applications [7,8,9,10,11].

Another important food property is friction, which has been investigated for applications in mastication [12,13,14] and robotic grasping [15]. In addition, 3D food geometry has been studied for food modeling [16,17], quality evaluation, and classification [18,19]. Food databases can also be found in the fields of food composition [20], nutrient profiles [21], food constituents, chemistry and biology [22], and food recognition [23,24,25,26]. However, most of the aforementioned food properties are difficult to apply in robotic food handling because they have not been investigated or measured in a robotic food handling scenario. Therefore, investigations and measurements specified for robotic food handling require increased attention and effort.

Researchers have also attempted to categorize food products and robotic end-effectors. Wurdemann et al. [27] proposed an approach to categorize food products for a food ordering process, and the key properties were considered as symmetry, surface condition, hardness, springiness, and resistance to damage. Erzincanli and Sharp [28] proposed a classification system for robotic food handling. Food products were classified based on their shape, dimension, surface, compliance, temperature, and weight. Robotic end-effectors were grouped based on their functions, such as clamping end-effectors, two-finger grippers, three (or more)-finger grippers, hard fingered end-effectors, and soft fingered end-effectors. A coding system was also created for food classification. Unfortunately, the aforementioned food classifications used qualitative descriptions of food properties, such as smooth and not smooth for the surface and rigid or hard and non-rigid or soft for compliance, which are insufficient for robotic food handling. Quantitative descriptions are required for better comparison and categorization.

Regarding robotic end-effectors, Fantoni et al. [29] wrote a detailed review of grasping devices for automated production processes. The authors summarized the grasping principles and detailed the end-effectors used in various applications, including in standard mechanical, electronic, and micro-assembly, in the food industry, in logistics, and in integrated grasping and processing. In recent years, soft robotic end-effectors have attracted attention and many applications in the food industry have been reported. Many soft grippers have the advantages of being lightweight, easy to control, adaptable to variations, and soft, which match the preferences of food handling applications. Consequently, soft robotic grippers are the most frequently commercialized soft robots. Examples include mGrip grippers [30], soft flexible grippers [31], modular-designed soft grippers [32], vacuum-driven soft grippers [33], and soft grippers [34].

In addition to commercialized soft end-effectors, there are various under-developed ones specialized for food handling tasks, such as the needle gripper for grasping salads [35], the soft suction gripper for handling fruits [36], the gripper using magnetorheological fluid for shape adaption [37], the soft gripper equipped with suction cups on each finger to improve its grasping capability [38], the origami-inspired soft gripper designed for grasping fragile food items [39], the soft jamming gripper for grasping various objects [40], the soft wrapping gripper for packaging chopped and granular foods [41], the dual-mode soft gripper combining grasping and suction for food packaging [42], and the finger-less soft gripper capable of generating multiple grasping modes [43]. A detailed review that introduces the principles of soft robotic grippers and their various applications can be found in [44].

However, despite the rapid development of soft end-effector technologies, their actual use in food production lines remains limited because of the large variety of food products and high-mix low-volume manufacturing features in the food industry. This is also one of the reasons it is difficult to select and design robotic end-effectors for food handling. In this study, we focused on addressing the difficulty of selecting appropriate robotic end-effectors for handling various food items. In addition, it was also expected to be useful for the investigation of new end-effector designs. Therefore, the main contributions of this study are as follows:1.Food categorization was proposed based on the food properties related to robotic handling.2.Pick-and-place tests were performed on the categorized food items using several commercialized and under-developed robotic end-effectors.3.A scoring system was proposed to evaluate the handling performance of robotic end-effectors.4.A visualization approach using a radar chart was proposed to present the evaluation results and compare of different end-effectors.

## 2. Materials and Methods

### 2.1. Concept

Consider a general food handling task in which a robotic manipulator equipped with a robotic end-effector is controlled to pick-and-place various food items, such as fruits, vegetables, and snacks. A robotic manipulator can be feasibly selected as long as its reach and payload satisfy the task requirements. However, selecting an appropriate robotic end-effector is difficult owing to the complexity and variety of handled food items. Furthermore, a proper method for evaluating the handling performance of robotic end-effectors is lacking.

Therefore, we propose an evaluation system for robotic effectors based on food and end-effector categorizations that aims to provide guidance for selecting an appropriate robotic end-effector for food handling tasks, as shown in Figure 1. First, food categorization is proposed based on a series of handling-related physical properties, such as weight, shape, friction coefficients, and Young’s modulus, as shown in Figure 1a. These physical properties are used to quantitatively classify different food items, which are helpful for evaluating the performance of robotic end-effectors in food handling tasks. The robotic end-effector categorization approach [45] is presented in Figure 1b. Robotic end-effectors are classified based on their contact positions. In Figure 1c, the scoring system used for the handling performance evaluation is illustrated. This scoring system is used to rate the handling performance of a robotic end-effector. Moreover, a visualization approach for the handling performance presentation, which displays the handling performance of robotic end-effectors using a radar chart, is proposed. The most efficient robotic end-effector was identified using the proposed evaluation system; this will be used for food handling tasks.

### 2.2. Food Categorization

Various physical properties have been proposed for classifying food items using traditional food categorization approaches. Food items can be comprehensively described and analyzed using multiple physical properties. However, this is difficult to consider, and it is not necessary to consider each physical property when conducting robotic food handling tasks. In this study, we mainly focus on a common food handling task and propose a new food categorization approach based on several handling-related physical properties, as described in Figure 1a. We considered two groups of handling-related physical properties: body and surface properties.

#### 2.2.1. Body Properties

Body properties describe the internal physical parameters of a food item, such as shape, size, weight, elasticity, viscosity, and fragility.

**Shape** describes the geometry of the body of a food item. Food items with a regular shape are convenient to handle, but irregularly shaped ones are less convenient to handle. In a food handling task, a robotic end-effector can intuitively determine whether a food item can be grasped based on shape information. For example, a clamping robotic end-effector is typically used for grasping food items with two parallel surfaces; however, it is rarely used for grasping food items without any parallel surfaces. In fact, it is difficult to determine the exact geometrical shape that matches a food item. Thus, a food item is considered to have a certain shape if it is roughly matched.

**Size** describes the two-dimensional magnitude of the body of a food item, which is commonly determined based on the grasping configuration in a food handling task. For example, a robotic end-effector is typically controlled to grasp the narrower sides of a fish rather than its wider sides. Therefore, the length of the grasping side of a food item is used to quantitatively determine its size.

**Weight** describes the heaviness or mass of a food item. In food handling tasks, the masses of grasped food items are different. There are light food items, such as a strawberry weighing approximately 20 g, and heavy food items, such as a watermelon weighing approximately 2000 g. Generally, most food items weigh up to 200 g.

**Elasticity and Viscosity** describe the ability of a food item to store and dissipate energy. They are coupled physical properties; thus, they are treated with elasticity and quantitatively described by the Young’s modulus in a food handling task. The elasticity of a food item affects its handling performance after being grasped.

**Fragility** represents the magnitude of a food item that resists the damage. The grasping force of a robotic end-effector should be properly determined to avoid damage when handling food items.

#### 2.2.2. Surface Properties

Surface properties describe the external physical parameters of a food item, such as friction, stickiness, temperature, and humidity.

**Friction** is a resisting force that prevents two objects from sliding freely against each other. As the lifting force of most robotic end-effectors is the friction force, friction is an important physical property related to food handling tasks. When considering stable grasping, static friction is of interest to us; therefore, we mainly focused on static friction in food handling tasks. Static friction can be quantitatively expressed as the maximum static friction coefficient, which is the ratio of the friction force divided by the normal force during grasping. The maximum static friction coefficient of a food item can be used to calculate the required lifting force for a robotic end-effector.

**Stickiness** is a food item’s ability to stick to a surface, characterized by the force of attraction between the surfaces. A food item with a sticky surface may stick to the robotic end-effector and affect the placing process. Therefore, stickiness is an important physical property in food handling tasks.

**Temperature** is a physical quantity that quantitatively expresses the perceptions of hotness and coldness. The surface of a food item has different physical properties if the food item it at different temperatures, for example, frozen meat and thawed meat.

**Humidity** is the amount of moisture presenting over the surface of a food item. Changes in humidity affect the friction coefficient of a food item, which indirectly influences food handling performance.

### 2.3. End-Effector Categorization

Many commercialized and under-researched robotic end-effectors have been developed for food handling tasks. The categorization of robotic end-effectors is necessary and useful for exploring their handling performance. In this study, we adopt the categorization method proposed by Wang in [45]. As shown in Figure 1b, robotic end-effectors are divided into six categories based on their contact position/positions when grasping a food item: (1) T type indicates that the end-effector handles a food item only at the top surface, such as the suction cups, Bernoulli principle-based grippers, adhesive end-effectors and needle grippers; (2) S type denotes that only the side surfaces of the food item are used during grasping and many two- or multi-fingered robotic end-effectors belong to this category; (3) B type indicates that the end-effector will be inserted under the food item for stable handling, such as the SWITL hand commercialized by FURUKAWA KIKO [46]; (4) TS type is a combination of T-type and S-type, and end-effectors belonging to this type handle food items using top and side surfaces simultaneously, such as the gripper commercialized by RightHand Robotics, Inc., which combines a suction cup and a three-fingered gripper [47]; (5) the BS type is a combination of B type and S type, and the end-effectors handle the food item using bottom and side surfaces simultaneously, such as the ones proposed by Ma et al. [48] and Gafer et al. [49]; (6) the TBS type is a combination of T, B, and S types, and the end-effectors belonging to this category tend to envelop the food item and use all available surfaces to stabilize the grasping action, such as the multi-functional gripper proposed by Sam and Nefti [50]. 

### 2.4. Scoring Approach

In the scoring approach, each food property is classified into different levels to obtain a score according to handling performance. In this study, seven food properties were considered during scoring: shape, size, weight, elasticity, fragility, friction, and stickiness.

#### 2.4.1. Shape Adaptability

The shapes of the food items are categorized into three levels: regular, semi-regular, and irregular. The regular level indicates that the food items have one of the following shapes: sphere, cylinder, ellipse, cube, torus, cuboid, or cone. The semi-regular level indicates that the food items have a shape in which most sides and interior angles are different but they have at least two parallel sides. The irregular level indicates that the food items have a shape in which all sides and interior angles are different. Consequently, shape adaptability is defined as the shape level that the robotic end-effector can successfully grasp and is scored as 1, 2, and 3 points for regular, semi-regular, and irregular levels, respectively. If the robotic end-effector fails to handle the food item for shape reasons, it receives a score of 0 points.

#### 2.4.2. Size Adaptability

Size is categorized into three levels, small, normal, and large, corresponding to sizes of 0–50 mm, 50–100 mm, and over 100 mm, respectively, which represents the two-dimensional magnitude of the body of the food item. Therefore, the size adaptability is scored as 1, 2, or 3 points for the capability of grasping small, normal, and large levels, respectively. If the robotic end-effector fails to grasp the food item because of size reasons, it receives a score of 0 points.

#### 2.4.3. Weight Adaptability

Weight is categorized into light, normal, and heavy levels, which indicates weight ranges of 0–50 g, 50–200 g, and over 200 g, respectively. Weight adaptability is scored as 1, 2, and 3 points for light, normal, and heavy grasping levels, respectively. If the robotic end-effector fails to grasp the food item for weight reasons, it receives a score of 0 points.

#### 2.4.4. Friction Adaptability

Friction is categorized into three levels based on the magnitude of the maximum static friction coefficient: low, medium, and high. Each level has a friction coefficient of 0–0.50, 0.50–1.00, and over 1.00 for low, medium, and high, respectively. In contrast to shape, size, and weight adaptability, friction adaptability is scored as 3, 2, and 1 points for the ability to cover low, medium, and high friction levels, respectively, to classify the ability to handle slippery items. If the robotic end-effector fails to grasp the food item owing to friction reasons, it receives a score of 0 points.

#### 2.4.5. Elasticity Adaptability

Elasticity is categorized into three levels, low, medium, and high, which correspond to Young’s moduli of 0–100 kPa, 100–500 kPa, and over 500 kPa. Consequently, elasticity adaptability is scored as 3, 2, and 1 points for adapting to low, medium, and high elasticity levels, respectively, to rate the ability to handle soft targets. If the robotic end-effector fails to grasp the food item owing to softness, it receives a score of 0 points.

#### 2.4.6. Stickiness Adaptability

During a pick-and-place operation, the stickiness of food items may be beneficial for grasping; however, it also significantly affects the positional accuracy of placement. Therefore, in this study, we directly use the position error after placement to evaluate the performance of different end-effectors. During the experiments, we set up a paper sheet with several concentric circles at predefined intervals. The center of the circles represents the desired placement position of the food item. The position error can then be measured after placement, and we define three stickiness levels as small, medium, and large, corresponding to position errors of 0–10 mm, 10–20 mm, and over 20 mm, respectively. Consequently, stickiness adaptability is scored as 3, 2, and 1 points for small, medium, and large levels, respectively. The stickiness adaptability score was zero if the end-effector failed to grasp the food item for other reasons.

#### 2.4.7. Fragility Adaptability

Fragility is an important measure of the resistance of food items to damage after placement, including rupture, collapse, and plastic deformation. During the pick-and-place operation, fragility is highly related to the grasping force applied by the end-effector. It is difficult to obtain the absolute grasping forces for soft end-effectors; thus, we used the control input (air pressure in the case of pneumatic end-effectors) as a criterion to evaluate fragility adaptability. In our experiments, three different air pressures were used: 60%, 80%, and 100% of the recommended pressures for positive-pressure-driven end-effectors. Pressures of −40 kPa, −60 kPa, and −80 kPa were used for vacuum-driven end-effectors. Therefore, we categorized fragility adaptability into low, medium, and high levels corresponding to the above-mentioned three air pressures. A high level indicates that the food item can be successfully handled without damage even at the highest air pressure. Consequently, the fragility adaptability is scored as 1, 2, and 3 points for low, medium, and high levels, respectively. A score of 0 means that the food item was damaged even when using the lowest air pressure. Similar to the stickiness adaptability, the end-effector receives a score of 0 if it fails to grasp the food item for other reasons.

### 2.5. Visualization Approach

Based on the performance scores, we propose an approach that uses a radar chart to visualize and compare the performances of different end-effectors, as shown in Figure 1c. The radar chart has seven axes, each representing the adaptability score of the end-effector for one food property. The overall performance of the end-effector forms a polygon in the radar chart. If one vertex of the polygon is located at the center point of the radar chart, it means the end-effector has a score of 0, which further suggests that the end-effector failed to handle the food item for reasons related to one specific food property. For example, in Figure 1c, we selected three end-effectors to handle an orange, and a performance comparison is displayed in the radar chart. The three colored lines represent the evaluation results for the corresponding robotic end-effector. The green end-effector has scores of [2, 3, 1, 3, 1, 2, and 3], which indicates that it can grasp a food item with a normal size (50–100 mm), heavy weight (over 200 g), regular shape, medium elasticity (100–500 kPa), low friction (0–0.5), medium stickiness (placement error of 10–20 mm), and high fragility. Using the radar-chart-based visualization approach, we can not only compare the performances of different end-effectors handling the same food item, but also compare the performances of one selected end-effector for handling different food items.

### 2.6. Experiment Methods

We conducted a series of food handling experiments to assess the performances of multiple soft robotic end-effectors using the proposed evaluation system. During the experiments, we controlled a four-degree-of-freedom (DOF) robotic arm (HSR-065A1, Denso Wave Inc., Aichi, Japan) with a robotic end-effector to pick up a food item from the picking plate and then place it on the placing plate, as shown in Figure 2a. A measurement scale was placed on the placing plate, the center of which indicated the desired placement position of the food item. The robotic arm was equipped with a 3D camera (RealSense D435i, Intel, Santa Clara, CA, USA) to determine the position and orientation of the food item. To drive the pneumatic end-effectors, an air compressor (SLP-15EFD, Anest Iwata Corp., Kanagawa, Japan) and pressure regulators (ITV2030, SMC Co., Japan) were used to generate desired air pressures. Solenoid valves were used to control the airflow and were switched by the IO interface of the robotic arm. The vacuum pressure was generated using an air ejector (VUH07-44A, PISCO Co., Ltd., Nagano, Japan). The entire robotic system was constructed using ROS 2.

#### 2.6.1. Tested Soft End-Effectors

Seven soft robotic end-effectors purchased and developed by our laboratory were used for the handling experiments: five commercialized and two under development. As shown in Figure 2b, the commercialized end-effectors consisted of a suction cup (T-type EE 2), two two-fingered grippers (S-type EE 4 and EE 5), a three-fingered end-effector (S-type EE 6), and a four-fingered gripper (S-type EE 7). The two end-effectors under development are a parallel gripper [51] (BS-type EE 1) and a bladder end-effector [52] (S-type EE 3).

#### 2.6.2. Tested Food Items

Considering the food categorization proposed in Section 2.2, 14 food items were selected for the handling experiments, as shown in Figure 2c, to cover as many food categories as possible. The 14 food items were a green pepper, a halved egg, an egg roll, a kamaboko slice, a hamburger, a fried shrimp, a strawberry, fried chicken, pasta, daifuku, a tomato, a boiled egg, a cucumber and a fish. To investigate the feasibility of using food samples instead of real food in handling experiments, commercialized food samples of the selected food items were also tested in the handling experiments. The handling-related physical properties of the real food items are listed in Table 1, where the size indicates the dimension in the grasping direction. The weight and size were measured using three real food items for each type of food; in particular, the size of pasta was determined using the size of the plate shown in Figure 2c. The elasticity (Young’s modulus) of all food items except pasta was measured using a motorized test stand (EMX-275, IMADA, Japan) with a curve fitting method [53]. We only conducted one time measurement to determine its order of magnitude as a reference, since the Young’s modulus of food items usually has large variations and individual differences. The friction was measured as the maximum static friction coefficient using a friction measuring machine (TL201 Tt, Trinity-Lab. Inc., Japan) with 5 measurements of one real food of each type except for pasta. The elasticity and friction coefficient of the pasta were obtained from [54,55], respectively.

#### 2.6.3. Experimental Protocols

The robotic system shown in Figure 2a was programmed to perform typical pick-and-place operations. The end-effectors was controlled to grasp the food item and lift it to a height of approximately 200 mm. The food item was then transported to a position approximately 150 mm above the place point. Finally, the end-effector moved downward to release the food item. The horizontal transport distance was approximately 540 mm. The duration of the entire pick-and-place circle, including four pauses between every pair of motions, was approximately 12 s. The air pressures for EE 1, EE 3, EE 4, and EE 7 were set to 60%, 80%, and 100% of their standard recommended pressures, respectively. For EE 2, EE 5, and EE 6, which were actuated by vacuum pressure, the experiments were conducted at three different vacuum pressures: approximately −40 kPa, −60 kPa, and −80 kPa. Five experimental trials were conducted for each food item, including both real food and the sample. Thus, the total number of experimental trials was 7 (EEs) × 14 (foods) × 2 (real and sample) × 3 (pressures) × 5 (trials)=2940 trials. A video demonstrating the experimental setups and some experimental trials can be found in the Appendix A.

## 3. Results and Discussion

Consider EE 5; the results are shown in Figure 3. T1, T2, and T3 represent the three different air pressures used in the experiments. Successful trials (indicated by green circles) indicated that the food item was stably picked up and placed at the desired location. Otherwise, the trial was considered a failure, and the reasons were divided into three different types: (1) grasping failure, in which the food item could not be picked up (red triangle); (2) damage failure, in which the food item was damaged (blue square); and (3) movement failure, in which the food item was dropped during the transporting motion (yellow diamond). Figure 3 shows that EE 5 succeeded in all trials for both the sample and real tomato, but failed 11 times in grasping the real egg roll owing to damage. In addition, EE 5 failed all trials in grasping the hamburger because it was large and it did not fit in the EE 5’s grasp. Interestingly, Figure 3 also shows that EE 5 succeeded in a few trials when handling real fish, but failed in all trials in grasping the fish sample owing to the heavy weight of the sample. The figure provides information of how the end-effector handles various food items.

Following the scoring approach proposed in Section 2.4, scores were calculated for each end-effector handling each food item. The results were visualized using a radar chart. Figure 4 shows the evaluation results of EE 5 when handling 14 real food items. We can see that EE 5 failed to handle kamoboko, hamburgers, daifuku, and fish for different reasons. For successful food items, EE 5 can cover a large variety of food properties, as shown in Figure 4o.

Figure 5 shows the evaluation results of the seven soft end-effectors for handling the same food item (real fried shrimp). We found that EE 4, EE 5, and EE 6 performed well in handling the fried shrimp. Furthermore, EE 1 and EE 7 performed better than EE 4, EE 5, and EE 6 because they had better adaptability to fragility. However, EE 2 and EE 3 failed to grasp the fried shrimp because of the shape of EE 2 and size of EE 3, as indicated in Figure 5b,c, with the corresponding vertices located at the center of the radar charts.

In addition, we propose an index to evaluate the overall performance of end-effectors and attempt to help select an appropriate end-effector for a given food item. The index was calculated as the production of all adaptability scores for a certain food item. We decided to use the production over summation because production can not only enlarge the differences among different end-effectors, but also can consider a score of 0 to clearly indicate a handling failure. Table 2 shows the calculated indices of the seven robotic end-effectors for handling 14 types of food items. For a given food item, the larger the index, the better the performance of the end-effector. End-effectors having the same index indicates that they performed similarly in handling the food item. A zero index indicates that the end-effector cannot handle the food item. From Table 2, we can easily see that only two end-effectors can handle pasta, hamburgers, fish, and cucumbers. No end-effector successfully handled a kamaboko piece. In contrast, many end-effectors handled fried shrimp, halved eggs, daifuku, and strawberries.

## 4. Conclusions

Owing to the large variety of food products, it is desirable for one end-effector to cover as many types of food items as possible. However, performing experimental tests on all food products is impractical. Therefore, it is beneficial to have an approach to evaluate end-effector performance and help in the selection and design of robotic end-effectors for food handling. In this study, we proposed an evaluation system consisting of food categorization based on handling-related physical properties, robotic end-effector categorization based on handling principles, a scoring system for performance evaluation, a visualization approach for display and comparison, and an ROS-2-based robotic system for conducting handling experiments.

To provide an example and demonstrate the evaluation process, we selected 14 real food items and their corresponding samples as handling targets. We selected seven pneumatic soft robotic end-effectors, both commercialized and under development, for the evaluation. After the experiments, the score of each end-effector was calculated and the performance was evaluated and visually displayed using radar charts for comparison. Through this process, we can not only obtain the overall performance of a robotic end-effector in handling various food items, but also compare the performances of different robotic end-effectors in handling the same food item. The overall performance of a robotic end-effector in handling various food items can quantitatively demonstrate its versatility and limitations, which may provide new ideas for modifications. However, a performance comparison of different end-effectors in handling the same food item can serve as a quantitative measure for selecting the proper end-effector for a certain handling task.

In this study, we chose 14 food items and seven pneumatic soft robotic end-effectors to handle evaluations as an example. In the future, more food products and robotic end-effectors will be tested and evaluated. The applicability of existing end-effectors will be continuously assessed. Meanwhile, new designs of soft robotic end-effectors will be derived and explored while guaranteeing food safety with appropriate materials and actuation methods. We will allow access to data including the food properties and experimental results as a reference, and we also encourage readers who are interested in performing the same evaluations on their end-effectors and food products to share their results if they prefer to do so. With the efforts of the research community, more food property data and evaluation results of robotic end-effectors can be gathered, and we believe the challenges of robotic food handling will be overcome.

## Figures and Tables

**Figure 1 foods-12-04062-f001:**
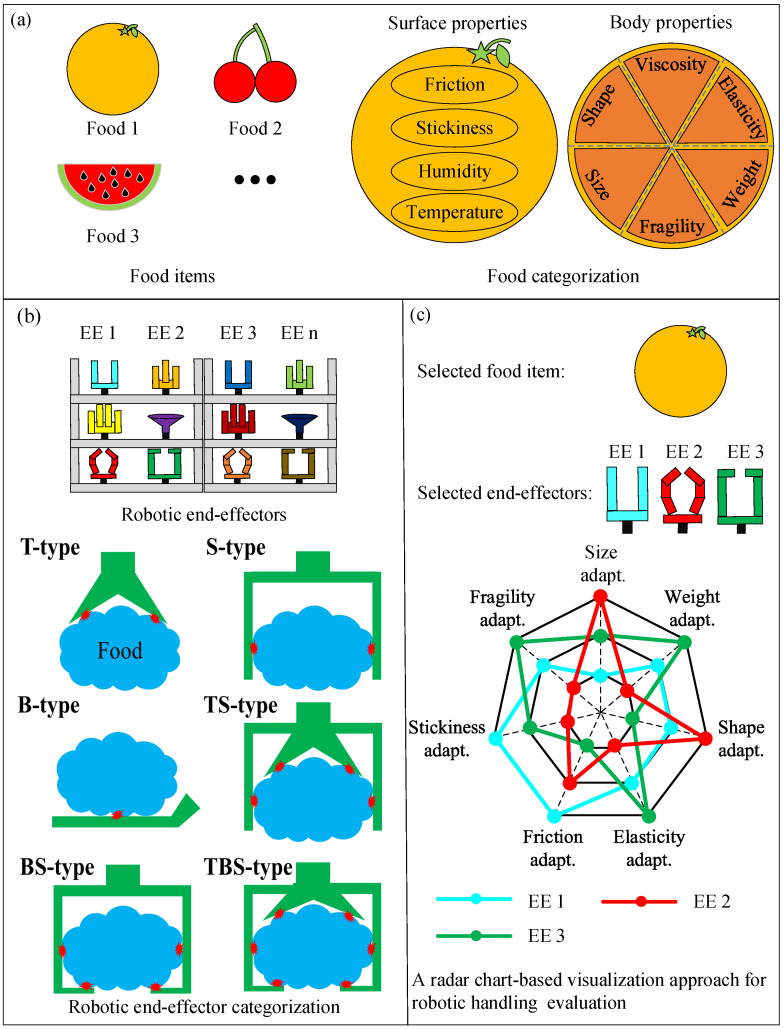
Food handling using the evaluation system of robotic end-effectors: (**a**) food categorization, (**b**) robotic end-effector categorization, and (**c**) food-categorization-based scoring and radar chart-based visualization approach for robotic handling evaluation.

**Figure 2 foods-12-04062-f002:**
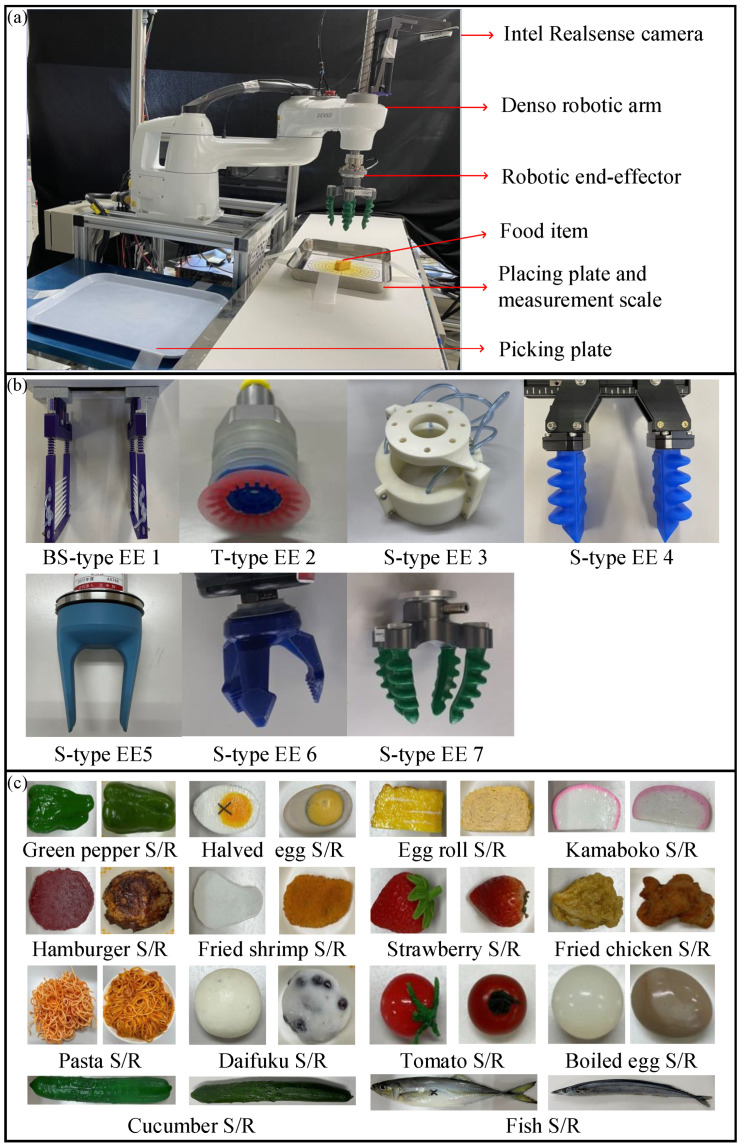
Task description of handling experiments: (**a**) experimental setup, (**b**) selected robotic end-effectors, and (**c**) selected food items with their corresponding physical twins (food samples).

**Figure 3 foods-12-04062-f003:**
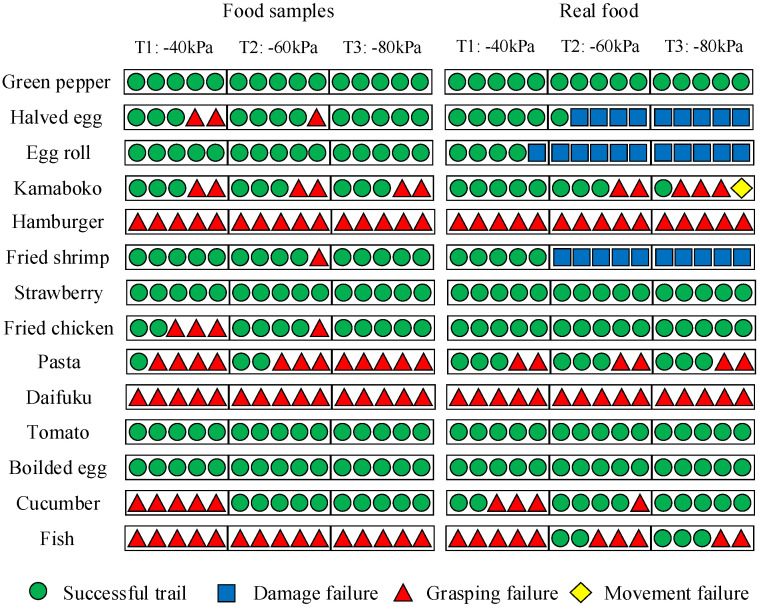
Results of handling experiments using EE 5. The green circle, blue square, red triangle, and yellow diamond indicate a successful trail, damage failure, grasping failure, and movement failure, respectively.

**Figure 4 foods-12-04062-f004:**
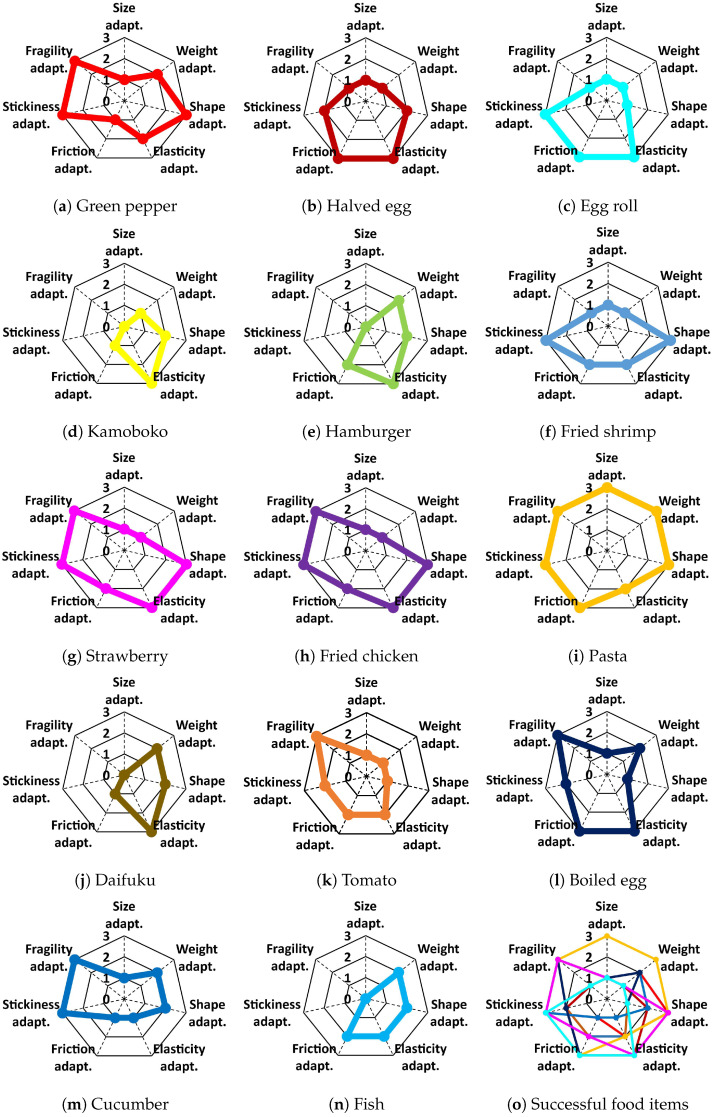
Evaluation results of EE 5. The handling performance related to each food item is displayed using a radar chart, and the right bottom radar chart shows the overall evaluation results on handling all fourteen food items.

**Figure 5 foods-12-04062-f005:**
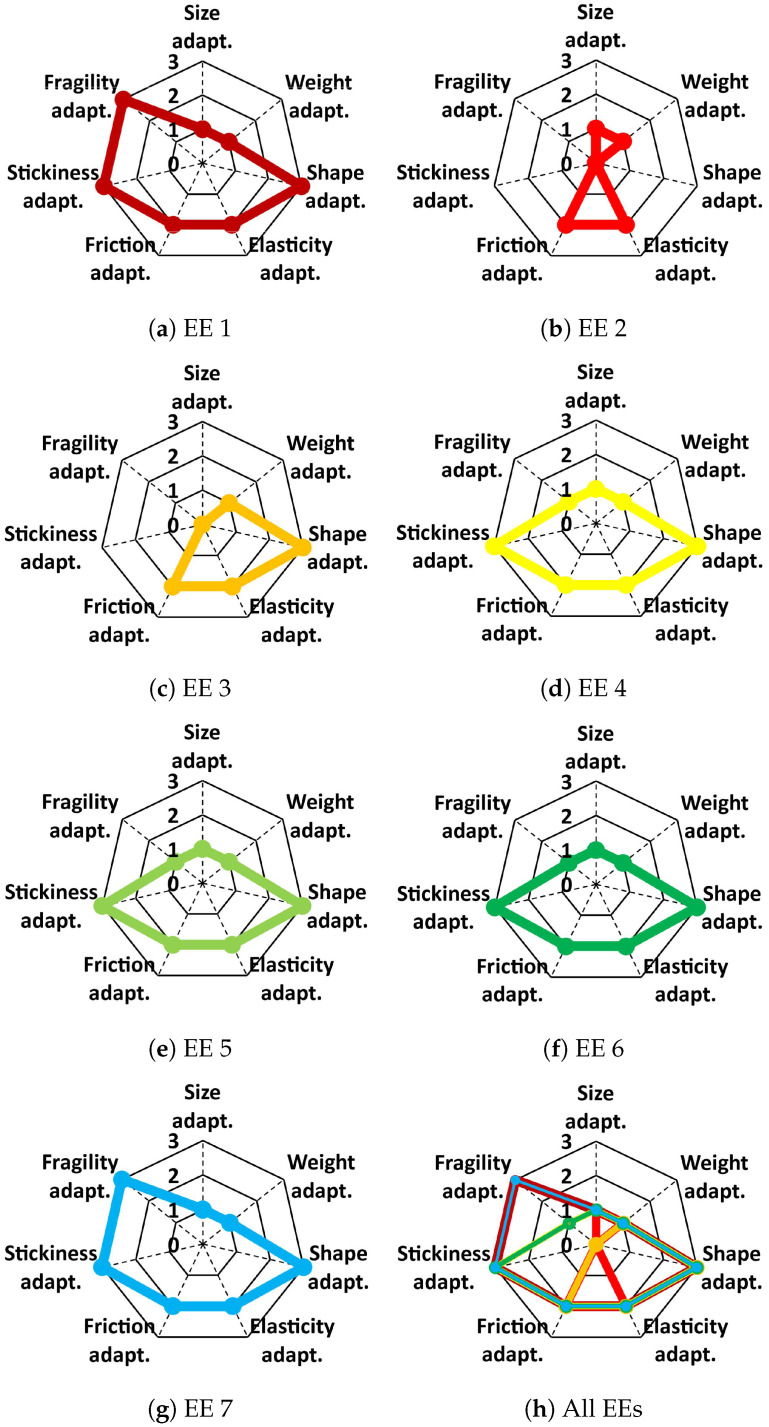
Evaluation results of seven robotic end-effectors in handling the same food item (real fried shrimp). The handling performance of each robotic end-effector is displayed using a radar chart, and the right bottom chart shows the comparison of the handling performances of all robotic end-effectors.

**Table 1 foods-12-04062-t001:** Handling-related physical properties of the fourteen real food items, where ’R’, ’SR’, and ’IR’ indicate regular, semi-regular, and irregular shapes, respectively.

Food Items	Weight (g)	Size (mm)	Shape	Elasticity (kPa)	Friction
Green pepper	33∼38	46∼54	IR	166.00	1.35∼1.53
Halved egg	21∼23	37∼39	SR	43.47	0.32∼0.46
Egg roll	25∼26	28∼29	R	25.11	0.41∼0.59
Kamaboko	5∼7	21∼22	SR	52.27	1.42∼1.68
Hamburger	139∼141	60∼67	SR	15.00	0.34∼0.42
Fried shrimp	22∼24	40∼41	IR	384.40	0.50∼0.72
Strawberry	14∼18	25∼28	IR	29.55	0.57∼0.95
Fried chicken	25∼32	38∼45	IR	19.55	0.42∼0.64
Pasta	309∼310	200	IR	158.00	0.20∼0.30
Daifuku	55∼62	44∼46	SR	11.15	1.54∼1.69
Tomato	12∼13	24∼26	R	410.30	0.44∼0.65
Boiled egg	41∼42	38∼40	R	52.46	0.32∼0.46
Cucumber	89∼129	20∼28	SR	987.20	1.02∼1.23
Fish	103∼108	35∼37	SR	138.70	0.60∼0.71

**Table 2 foods-12-04062-t002:** Evaluation results of seven robotic end-effectors: the weight is calculated using the product of the score of each handling-related property of a food item.

	EE 1	EE 2	EE 3	EE 4	EE 5	EE 6	EE 7
Green pepper	108	0	0	0	108	72	108
Halved egg	108	0	0	108	36	108	108
Egg roll	0	0	0	27	27	54	0
Kamaboko	0	0	0	0	0	0	0
Hamburger	432	0	0	432	0	0	0
Fried shrimp	108	0	0	36	36	36	108
Strawberry	162	0	0	162	162	162	162
Fried chicken	162	0	0	0	162	162	162
Pasta	0	0	0	0	1458	1458	0
Daifuku	216	0	216	216	0	216	216
Tomato	24	0	0	0	24	24	24
Boiled egg	54	0	0	0	108	108	108
Cucumber	24	0	0	0	36	0	0
Fish	144	0	0	144	0	0	0

## Data Availability

The data presented in this study are available on request from the corresponding author.

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
