# Peer review of "An Evaluation System of Robotic End-Effectors for Food Handling"

_foods, 2023, doi:10.3390/foods12224062_

Round 1

Reviewer 1 Report

Comments and Suggestions for Authors

In this study the authors evaluated a system for robotic end-effectors based on food and end-effector categorizations. The authors chose  food items and seven pneumatic soft robotic end-effectors to handle evaluations as an example. 

The main contributions of this study are as follows: 

1. Food categorization was proposed based on the food properties related to robotic handling. 

2. Pick-and-place tests were performed on the categorized food items using several commercialized and under-developed robotic end-effectors. 

3. A scoring system was proposed to evaluate the performance of robotic end-effectors. 

4. A visualization approach using a radar chart was proposed to present the evaluation results and compare of different end-effectors.

The authors will open-source the data including the food properties and experimental results as a reference, and we will also encourage readers who are interested in performing the same evaluations on their end-effectors and food products to share their results if they prefer to do so.

The paper is well done but I have some remarks:

- the authors should edit the radar charts (figure 4-5)

- the authors should report the statistic value of the physical properties of the real food in table 1

Comments on the Quality of English Language

The english language is good

Author Response

Thank you very much for your valuable time and constructive comments. We have revised our manuscript according to your comments as detailed below. In this document, we intend to answer the questions and concerns raised by you and indicate the changes we have made corresponding to your comments. Through this document, letter “C” indicates comment or concern from you, and letter “R” indicates the corresponding reply and changes. All changes were colored in red in the revised manuscript.

In this study the authors evaluated a system for robotic end-effectors based on food and end-effector categorizations. The authors chose food items and seven pneumatic soft robotic end-effectors to handle evaluations as an example.

The main contributions of this study are as follows:

  1. Food categorization was proposed based on the food properties related to robotic handling.
  2. Pick-and-place tests were performed on the categorized food items using several commercialized and under-developed robotic end-effectors.
  3. A scoring system was proposed to evaluate the performance of robotic end-effectors.
  4. A visualization approach using a radar chart was proposed to present the evaluation results and compare of different end-effectors.

The authors will open-source the data including the food properties and experimental results as a reference, and we will also encourage readers who are interested in performing the same evaluations on their end-effectors and food products to share their results if they prefer to do so.

The paper is well done but I have some remarks:

C1: the authors should edit the radar charts (figure 4-5)

R1: Thank you very much for your comment and suggestion.

To increase the readability and appearance, we have made the following modifications to figures 4 and 5:

(1) The subtitle of each subfigure is centered.

(2) Each subfigure size was slightly enlarged, and thus it would be more explicit.

Modifications in the manuscript: page 12, figure 4, and page 13, figure 5.

C2: the authors should report the statistic value of the physical properties of the real food in table 1

R2: Thank you very much for your comment and suggestion. To response to your comments, we have made the following modifications to Table 1.

(1) We added statistic values (range of the measured values) of weight and size of each food in Table 1, which were measured using three real food items for each type of food.

(2) We also added statistic values (range of the measured values) of the static friction of each food in Table 1. The measurement experiments were conducted 5 times for each food.

(3) Since the Young’s modulus of food items usually has large variations and individual differences, we only conducted one time experiment to determine its order of magnitude as a reference.

Modifications in the manuscript: page 9, Table 1, and lines 346-354.

Reviewer 2 Report

Comments and Suggestions for Authors

Dear Authors, below are detailed comments on the manuscript:

1) Add the work goal in "Abstract",

2) Both in "Introdaction" and in "Discussion" refer more broadly to food safety (in the context of the proposed solution),

3) Lines 97-105 - indicate the main research objective and additional objectives,

4) Chapter No. 5. "Conclusions and discussions" should, in my opinion, be separated into "Discussions" and "Conclusions" (or "Summary").

Author Response

Thank you very much for your valuable time and constructive comments. We have revised our manuscript according to your comments as detailed below. In this document, we intend to answer the questions and concerns raised by you and indicate the changes we have made corresponding to your comments. Through this document, letter “C” indicates comment or concern from you, and letter “R” indicates the corresponding reply and changes. All changes were colored in red in the revised manuscript.

Dear Authors, below are detailed comments on the manuscript:

C1: Add the work goal in "Abstract"

R1: Thank you very much for your useful comment and suggestion.

In Abstract, we added “Our primary goal is to maximize the applicability of existing end-effectors and efficiently develop novel ones, and therefore, it is necessary to categorize food products and end-effectors from the viewpoint of robotic handling and establish their correspondence through an effective evaluation approach.”

Modifications in the manuscript: page 1, lines 3-6.

C2: Both in "Introduction" and in "Discussion" refer more broadly to food safety (in the context of the proposed solution)

R2: Thank you very much for the useful comment and suggestion.

In Introduction, we added “Food contamination is another consistent challenge facing by the food industry worldwide, as demonstrated by multiple recalls due to poor food quality, undeclared allergens, and other forms of contamination. To overcome global labor shortages and promote food safety in food industry, the use of robotics and automation has drawn significant attention from both academia and industry.”

Modifications in the manuscript: page 1, lines 24-28.

In Conclusions, we added “The applicability of existing end-effectors will be continuously assessed. Meanwhile, new designs of soft robotic end-effectors will be inspired and explored while guaranteeing food safety problems with appropriate materials and actuation methods.”

Modifications in the manuscript: page 14, lines 438-441.

C3: Lines 97-105 - indicate the main research objective and additional objectives

R3: Thank you very much for the suggestion.

We added “In this study, we focused on addressing the difficulty of selecting appropriate robotic end-effectors for handling various food items. In addition, it was also expected to be useful for the investigation of new end-effector designs.” to explain our main research objective and additional objectives.

Modifications in the manuscript: page 3, lines 98-101.

C4: Chapter No. 5. "Conclusions and discussions" should, in my opinion, be separated into "Discussions" and "Conclusions" (or "Summary").

R4: Thank you very much for the comment and suggestion.

Considering this comment and a similar comment from another reviewer, we changed the structure of the manuscript and the titles of many sections. Section 2 is now entitled as “Materials and Methods”, Section 3 became “Results and Discussion”, and Section 4 is now “Conclusions”.

The discussions are now conducted in the Section 3 “Results and Discussion”. In Section 4 “Conclusions”, we made a summary of the manuscript and suggested some future works. 

Modifications in the manuscript: section titles of Sections 2 to 4.

Reviewer 3 Report

Comments and Suggestions for Authors

General remark

This manuscript reports evaluation system of robotic end-effectors for food handling. This manuscript is well-described. It could be used as fundamental information for application of robotic end-effectors for food handling. However, some issues need to be addressed before acceptance as follows;

1.    The manuscript is well-described; however, format of section in manuscript should be organized ex. Materials and Methods, Results and Discussion etc.

2.    Section 3.3.3 Size adaptability: Is it 2 or 3 dimension? Please clarify.

3.    How authors propose these 7 types of robotic end-effectors?

4.    Table 1 and 2: How many replicates in each trial?

Author Response

Thank you very much for your valuable time and constructive comments. We have revised our manuscript according to your comments as detailed below. In this document, we intend to answer the questions and concerns raised by you and indicate the changes we have made corresponding to your comments. Through this document, letter “C” indicates comment or concern from you, and letter “R” indicates the corresponding reply and changes. All changes were colored in red in the revised manuscript.

This manuscript reports evaluation system of robotic end-effectors for food handling. This manuscript is well-described. It could be used as fundamental information for application of robotic end-effectors for food handling. However, some issues need to be addressed before acceptance as follows.

C1: The manuscript is well-described; however, format of section in manuscript should be organized ex. Materials and Methods, Results and Discussion etc.

R1: Thank you very much for the comment and suggestion.

According to the comment and a similar comment from another reviewer, we have changed the structure of the manuscript and the titles of most sections. Section 2 is now “Materials and Methods”, Section 3 became “Results and Discussion”, and Section 4 is “Conclusions”.

Modifications in the manuscript: section titles of Sections 2 to 4.

C2: Section 3.3.3 Size adaptability: Is it 2 or 3 dimensions? Please clarify.

R2: Thank you very much for the question.

The size adaptability is a two-dimensional concept. We added “Size is categorized into three levels: small, normal, and large, corresponding to sizes of 0-50 mm, 50-100 mm, and over 100 mm, respectively, which represents a two-dimensional magnitude of the body of the food item.” to clarify this information. In Section 2.2.1 “Body Properties”, the size is pre-defined as a two-dimensional magnitude of the body of a food item, which is commonly determined based on the grasping configuration in a food handling task.

Modifications in the manuscript: page 6, lines 237-238.

C3: How authors propose these 7 types of robotic end-effectors?

R3: Thank you very much for question.

First, we consider both commercialized and under-developed robotic end-effectors. For commercialized ones, we choose the ones that we already have in hand. For the under-developed ones, we choose the ones that developed by ourselves and had relatively good performances. Second, according to our end-effector categorization method, these seven end-effectors belong to three types, which could be representative for robotic handing tasks. In the future, more end-effectors of different types will be evaluated.

We added “Seven soft robotic end-effectors that purchased and developed by our laboratory were used for the handling experiments: five commercialized and two under-developed.”

Modifications in the manuscript: page 9, line 329.

C4: Table 1 and 2: How many replicates in each trial?

R4: Thank you very much for the question.

Based on our understanding, “each trial” may have two meanings. One is to obtain the values in Tables 1 and 2, how many experiments we conducted. The other is to treat Table 1 or Table 2 itself as a trial.

Consider the first case, for Table 1, the size and weight values were measured by three real food of each type of food, the Young’s Modulus of each food was conducted one time experiment to determine its order of magnitude as a reference because Young’s Modulus usually has large variations and individual differences, and the static friction of each food was conducted 5 times experiments using one real food. For Table 2, we consider real food and corresponding samples for handling experiments. For each food item, we conducted five experimental trials. Therefore, total experimental trials were 7 (num. of EEs) â…¹14(num. of food items) â…¹2 (real and sample) â…¹ 3 (num. of pressure values) â…¹5 (trials)=2940 trials.

Consider the second case, Table 1 or Table 2 itself as a trial. We only conducted one trial because the entire environment was time- and resource-consuming. In the future, we plan to try more robotic end-effectors and food items, but our own efforts are still limited. With the efforts of the research communities, more data of food properties and evaluation results of robotic end-effectors would be accumulated, and we believe the challenges of robotic food handling will be overcome.

Modifications in the manuscript: page 9, Table 1, lines 346-354, and page 14, lines 444-446.